# Uric Acid Reacts with Peroxidasin, Decreases Collagen IV Crosslink, Impairs Human Endothelial Cell Migration and Adhesion

**DOI:** 10.3390/antiox11061117

**Published:** 2022-06-04

**Authors:** Bianca Dempsey, Litiele Cezar Cruz, Marcela Franco Mineiro, Railmara Pereira da Silva, Flavia Carla Meotti

**Affiliations:** Department of Biochemistry, Institute of Chemistry, University of São Paulo, Av. Prof Lineu Prestes, 748. Office 1004, São Paulo 05508-000, Brazil; bianca.dempsey@usp.br (B.D.); litieleccruz@gmail.com (L.C.C.); marcelamineiro@gmail.com (M.F.M.); railmara@iq.usp.br (R.P.d.S.)

**Keywords:** peroxidasin, uric acid, oxidation, HUVEC, extracellular matrix, collagen IV, crosslink

## Abstract

Uric acid is considered the main substrate for peroxidases in plasma. The oxidation of uric acid by human peroxidases generates urate free radical and urate hydroperoxide, which might affect endothelial function and explain, at least in part, the harmful effects of uric acid on the vascular system. Peroxidasin (PXDN), the most recent heme-peroxidase described in humans, catalyzes the formation of hypobromous acid, which mediates collagen IV crosslinks in the extracellular matrix. This enzyme has gained increasing scientific interest since it is associated with cardiovascular disease, cancer, and renal fibrosis. The main objective here was to investigate whether uric acid would react with PXDN and compromise the function of the enzyme in human endothelial cells. Urate decreased Amplex Red oxidation and brominating activity in the extracellular matrix (ECM) from HEK293/PXDN overexpressing cells and in the secretome of HUVECs. Parallelly, urate was oxidized to 5-hydroxyisourate. It also decreased collagen IV crosslink in isolated ECM from PFHR9 cells. Urate, the PXDN inhibitor phloroglucinol, and the PXDN knockdown impaired migration and adhesion of HUVECs. These results demonstrated that uric acid can affect extracellular matrix formation by competing for PXDN. The oxidation of uric acid by PXDN is likely a relevant mechanism in the endothelial dysfunction related to this metabolite.

## 1. Introduction

Uric acid is the end product of purine metabolism in humans since we do not express urate oxidase. Uric acid accumulates in plasma as the mono-anion urate (pKa 5.4), and the plasma concentration varies between 50–400 µM in normouricemia, reaching millimolar concentration in hyperuricemia [1,2]. Due to the high concentration in plasma and the low one-electron reduction potential (*E°* = 0.56 V, pH 7.0, HU^●−^, H^+^/UH_2_^−^) [3], urate is considered the main antioxidant in this fluid. It is a facile electron donor, capable of chelating transition metals and scavenger oxidants such as the hydroxyl radical and singlet oxygen [2,4]. Despite its effects against oxidizing agents, uric acid can act as a DAMP (damage-associated molecular pattern) [5], which activates the inflammasome cascade [6,7,8]. 

Urate levels are positively associated with cardiovascular diseases [9,10,11,12], and several mechanisms point to a causal role for urate. This metabolite increases oxidative stress, decreases the availability of nitric oxide, causes endothelial dysfunction, promotes inflammation, upregulates the angiotensin system, and induces proliferation and migration of vascular smooth muscle cells [13,14,15,16,17,18].

We and others demonstrated that urate is a substrate for the inflammatory peroxidases myeloperoxidase and lactoperoxidase [19,20]. The products urate free radical and urate hydroperoxide are unstable and potentially harmful by inducing redox imbalance in favor of a pro-oxidant environment [18,21,22]. Of relevance, urate has been considered the main organic substrate for peroxidases in human plasma [23]. Therefore, we investigated whether urate would also be a relevant substrate for the heme-peroxidase, peroxidasin 1 (PXDN1 or hsPxd01), an extracellular enzyme abundantly expressed in several tissues [24]. The oxidation of urate by PXDN could promote oxidative stress by the formation of urate-derived electrophiles and disrupt extracellular matrix formation, which is the main function of this enzyme. PXDN shares structural features with myeloperoxidase and lactoperoxidase [25,26], and all of them oxidize halides or pseudo-halides to produce the respective hypohalous or pseudo-hypohalous acids [27,28,29]. PXDN contains a large N-terminal leucine-rich repeat domain, four immunoglobulin-like domains, and a VWC (von Willebrand factor type C) domain in the C-terminal portion [26,30,31]. The enzyme catalyzes the formation of hypobromous acid (HOBr) that further oxidizes methionine, lysine, or hydroxylysine residues to form covalent sulfilimine bonds in collagen IV [32]. The crosslink between the NC1 domains of two adjacent collagen IV protomers structures the basement membrane in the extracellular matrix [33]. 

A recent paper showed that urate can be oxidized by a truncated fully active PXDN (hsPxd01-con4). The rate constant of the reaction of compound I hsPxd01-con4 with urate (1.9 × 10^4^ M^−1^s^−1^) was around two orders of magnitude lower than that with bromide (5.6 × 10^6^ M^−1^s^−1^) [28,34]. However, urate can be a significant competitor against bromide considering their plasma concentrations, 50–400 µM and 60–75 µM, respectively [35], being even more relevant in hyperuricemic patients. In addition, the reaction of compound II hsPxd01-con4 with urate was slow (5.8 × 10^2^ M^−1^s^−1^), revealing that the complete turnover of the enzyme may be slowed down in the presence of urate. In this sense, urate would rather act as an inhibitor of the enzyme than a substrate, but independently of the exact mechanism, it would disturb basement membrane formation.

Considering the important role of PXDN in extracellular matrix formation and vascular endothelial homeostasis, we hypothesized that the oxidation of uric acid by PXDN could contribute to vascular disease by two mechanisms: (1) the oxidation of uric acid could divert the production of HOBr and collagen IV crosslink by the enzyme; (2) the production of urate free radical and its electrophilic-derived products could contribute to oxidative stress and alter the function of proteins that are targets for oxidation/uratylation [21,22,36]. Because urate has well-known effects on endothelial dysfunction, atherosclerosis, and cardiovascular diseases [9,11,37,38], we wondered whether the reaction of urate with PXDN could be one of the mechanisms for the harmful effect of uric acid in vascular tissue. 

In this study, we extended previous findings on the reaction of PXDN with urate [34,39] by investigating more deeply the consequences of urate oxidation by PXDN, including how it can affect HOBr production and extracellular matrix formation. We initially identified PXDN as a peroxidase present in the secretome of vascular endothelial cells, HUVECs. The products of urate oxidation and the effects of urate upon peroxidase and bromination activity of PXDN were investigated using HUVECs and HEK293 cells overexpressing PXDN. The effects of urate upon the crosslink were assayed with isolated ECM of PFHR9 cells, a cell line that overproduces ECM. Finally, we showed the effects of uric acid, as well as PXDN inhibition, on the migration and adhesion of HUVECs.

## 2. Experimental Procedures

### 2.1. Reagents and Antibodies

RPMI medium and fetal bovine serum (FBS) were purchased from Vitrocell (Campinas, Brazil). Amplex™ Red Reagent, Lipofectamine™ RNAiMAX Transfection Reagent Lipofectamine™ 3000 Transfection Reagent were obtained from Invitrogen (Life Technologies, Waltham, MA, USA). Solvents were purchased from JTBaker (Thermo Fisher Scientific, Life Technologies, Waltham, MA, USA). Mass Spectrometry Grade Trypsin was purchased from Promega (Madison, WI, USA). Iodoacetamide and dithiothreitol (DTT) were obtained from Bio-Rad Laboratories (Hercules, CA, USA). 4-aminobenzoic acid hydrazide (ABAH) was purchased from Fluka Chemicals (Seelze, Hanover, Germany). Amicon^®^ Ultra-15 mL and Amicon^®^ Ultra-0.5 mL centrifugal filters 10 kDa cutoff were acquired from Merck Millipore (Merck KGaA, Darmstadt, Germany). Plasmids (vector carrying pcDNA3.1) were kindly provided by Prof. Geiszt Miklós from Semmelweis University (Budapest, Hungary). Anti-COL4A2 and Anti-V5 antibodies were purchased from Invitrogen (Life Technologies, Waltham, MA, USA), anti-rabbit IgG HRP-linked Antibody from Sigma Aldrich (Merck KGaA, Darmstadt, Germany) and Collagenase 1 from Gibco (Life Technologies, Waltham, MA, USA). Small interfering (si) RNAs targeting PXDN (siPXDN) and control siRNA (siRNA scramble) were obtained from Invitrogen (Life Technologies, Waltham, MA, USA). All other reagents were purchased from Sigma Aldrich (Merck KGaA, Darmstadt, Germany). HOBr was prepared by adding an equal volume of 50 mM HOCl to 250 mM sodium bromide in phosphate buffer 50 mM pH 7.4 while vortexing. After 10 min, HOBr concentration was determined by measuring absorbance at 329 nm with ε_HOBr_ = 335 M^−1^cm^−1^ [39] after 1:100 dilution in 500 mM NaOH. 

### 2.2. Cell Culture and Transfection

A selection of immortalized human umbilical vein endothelial cell line (HUVEC) was kindly provided by Prof. Francisco Laurindo’s group from INCOR (São Paulo—Brazil). HUVECs were maintained in RPMI medium containing 10% fetal bovine serum (FBS), streptomycin (100 mg/mL), and penicillin (30 mg/mL) at 37 °C in a 5% CO_2_ atmosphere. For experiments, confluent adherent cells were harvested using a solution containing trypsin (0.1%) and EDTA (0.5 mM) prepared in phosphate-buffered saline solution (PBS; 10 mM, pH 7.4). After centrifugation and counting, cells were seeded on culture plates in the appropriate concentration. HUVECs were used between passages 6 and 10. Human Embryonic Kidney cells (HEK293 line - ATCC, Manassas, VA, USA, CRL-1573) and PFHR9 (ATCC, Manassas, VA, USA, CRL-2423), an epithelial cell from mouse embryonal carcinoma, were maintained in high glucose DMEM (Dulbecco’s Modified Eagle’s Medium) containing 10% fetal bovine serum (FBS), streptomycin (100 mg/mL) and penicillin (30 mg/mL) at 37 °C in a 5% CO_2_ atmosphere. For experiments, confluent adherent cells were harvested using a solution containing trypsin (0.1%) and EDTA (0.5 mM) for HEK293 cells and trypsin (0.25%) and EDTA (0.53 mM) for PFHR9 cells. The plasmids (vector carrying pcDNA3.1/PXDN-V5-His) were transfected into HEK293 cells using lipofectamine 3000 Transfection Kit according to the manufacturer’s instructions. After transfection, cells were treated for 4 weeks with 500 μg/mL of geneticin (G418) as a selection antibiotic. PXDN expression was confirmed by Western blot. Briefly, 20 µg HEK293/PDXN lysate was separated on a 10% reducing SDS-polyacrylamide gel and transferred to a polyvinylidene difluoride (PVDF) membrane. Membranes were incubated with primary antibody to Anti-V5 (1:10,000) and subsequently incubated with HRP-conjugated secondary antibody (anti-mouse—1:10,000) to detect protein expression by chemiluminescence. 

### 2.3. siRNA Transfection 

HEK293 overexpressing PXDN were seeded at 50% confluence and transfected with 50 or 100 nM siRNA PXDN (5′-CCUCCAUCCUAGAUCUUCGCUUUAA-3′) or siRNA control/scramble (5′-CCUCCCUCAUAGAUGUUCCCUUUAA-3′) using lipofectamine RNAiMAX Transfection Reagent according to the manufacturer’s instructions. After 24 h, V5 expression was evaluated. Sequences of the siRNAs were provided as described by Péterfi et al. (2009) [24]. As the concentration of 50 nM produced a significant inhibition (Appendix A), this concentration was selected for the experiments in HUVECs. HUVECs were transfected as described above, and after 24 h, the transfection medium was changed to a modified RPMI medium (without phenol red and FBS). After 90 min, the secretome was concentrated and peroxidase activity was assayed by Amplex Red. In the adhesion assay, cells that had been transfected by 24 h were harvested and seeded (2.5 × 10^5^ cells/well) for a further 24 h in 96-well plates without coating in RPMI medium 0.2% BSA. In the migration assay, 24 h after transfection, a scratch was made in the cell monolayer, and cells were washed with PBS and incubated with RPMI medium 0.2% BSA for 24 h. 

### 2.4. Isolation of Secretome and Extracellular Matrix 

In summary, cells were grown until 80% confluence in 100 mm culture plates. To isolate secretome, cells were washed twice with PBS (10 mM, pH 7.4) and incubated for 1 h 30 min (HUVEC), or 36 h (HEK293) with culture media without FBS and phenol red. The culture media were collected after the incubation period, centrifuged at 1400 rpm for 10 min, and filtered through disposable 0.22 μm filters to remove cellular debris and floating cells. Next, the secretome was concentrated 200-fold on Amicon-Ultra 15 mL–10 kDa filters. The ECM isolation procedure was followed as described by Bathish et al. (2018) [39]. Briefly, cells were lysed with 1% sodium deoxycholate and 1 mM EDTA in 50 mM Tris-HCl pH 8.0, scraped, sonicated, and spun. For some experiments, the lysate portion (soluble fraction) was collected at this step. The precipitated ECM was washed twice with 1 M NaCl 50 mM Tris-HCl pH 7.4, suspended in phosphate buffer 50 mM, pH 7.4, and sonicated again to homogenize. Total protein content in the isolated portions was quantified by Pierce™ BCA Protein Assay Kit (Thermo Fisher Scientific, Life Technologies, Waltham, MA, USA).

### 2.5. Amplex Red Assay

To measure peroxidase activity, 50 μM Amplex red and 50 μM H_2_O_2_ were added to 10 μg total protein content from cellular fractions (lysate, secretome, and ECM) to a final volume of 200 μL (phosphate buffer, 50 mM pH 7.4) into 96-well fluorescence microplates. Fluorescence (λ_ex_ = 540 nm, λ_em_ = 590 nm) was recorded every 30 s for 90 min in a microplate reader (Biotek Synergy, Biotek, Winooski, VT, USA). The results were calculated within the linear fluorescence increase and expressed as the rate of Amplex Red oxidation per minute. 

### 2.6. Quantification of Bromo-Tyrosine by HPLC

A total of 100 µg protein from cellular fractions was incubated with L-Tyrosine 1 mM, 100 mM NaBr, and 100 µM H_2_O_2_ for 4 h at 37 °C in a thermoblock (Eppendorf -Hamburg, Germany). Samples were injected into a Prominence HPLC instrument (Shimadzu, Kyoto, Japan). The products 3-bromotyrosine and 3,5-dibromotyrosine were separated from free L-tyrosine in a Shim-pack VP-ODS reverse phase column (5 µm resin, 4.6 mm × 250 mm—Shimadzu, Kyoto, Japan) equilibrated with 100% solvent A (0.1% trifluoroacetic acid, pH 2.5). L-Tyrosine and its brominated products were eluted at a flow rate of 1 mL/min with a linear gradient generated with solvent B (0.1% trifluoroacetic acid in methanol, pH 2.5) as follows: 0% solvent B for 5 min, 0% to 100% solvent B over 30 min, and 100% solvent B for 10 min. 3-Bromotyrosine and 3,5-dibromotyrosine were monitored on a diode array detector and quantified at 280 nm against a standard curve constructed with synthetized HOBr and L-Tyrosine [40]. 

### 2.7. Detection of Urate Oxidation Products by LC-MS/MS

In total, 100 µg total protein from HUVEC secretome or HEK293 cells ECM was incubated with urate (100 and 200 µM) and H_2_O_2_ (100 µM) to a final volume of 200 µL (50 mM phosphate buffer, pH 7.4) for 30 min at 37 °C. After incubation, samples were diluted in 2% trichloroacetic acid (TCA) and 60% acetonitrile (ACN) containing 16.6 μM uric acid 1,3-^15^N_2_ as internal standard. Samples were centrifuged at 20,000× *g* (2 min, 4 °C). The supernatant was filtered (PTFE filter 0.22 µm) and injected into the LC system coupled to a Q-TOF mass spectrometer (6600 Triple-TOF; AB Sciex), electrospray ionization source (ESI), operating in the negative mode, as described before [18] with modifications. The fragments were obtained through collision energy (−20 V) and −80 V of DP (declustering potential) for all analytes. The source temperature was 450 °C, and the spray voltage was adjusted to 4500 V. The chromatographic method was developed on a UPLC system from Nexera (Shimadzu, Kyoto, Japan) using a UPLC BEH Amide 1.7 μM (2.1 × 100 mm, Acquity Waters, Waters Corporation, Milford, MA, USA). The mobile phase was 10 mM ammonium acetate pH 6.8 (solvent A) and acetonitrile (solvent B). The separation was carried out in a gradient mode: 0–1 min: 90% B; 1–3 min: 50% B; 3–11 min: 50% B; 11–12 min: 2% B, 11–23 min: 2% B and 23–35 min: 90% B with a flow rate of 0.2 mL/min and an injection volume of 10 µL, the column was maintained at 25 °C. The quantification of hydroxyisourate was corrected by the internal standard uric acid 1,3-^15^N_2_. The mass transitions used were *m/z* 167.0160→126.0201 for uric acid, *m/z* 169.0151→127.0201 uric acid 1,3-^15^N_2_, and *m/z* 183.0108→140.0102 for hydroxyisourate (HOU) and plotted against a standard curve. The fragmentation of hydroxyisourate from a standard solution as well as its retention time was used to confirm hydroxyisourate detection in the samples.

### 2.8. Collagen IV Crosslink Assay

PFHR9 cells were cultured in DMEM as described above until confluence. Then, the medium was daily changed with the addition of 50 μM PHG (phloroglucinol), a heme peroxidase inhibitor, for 6 days for matrix deposition [39,41]. ECM isolation was conducted as described above for HUVEC and HEK293/PXDN cells. The final precipitate was sonicated and suspended in PBS. For the treatments, 750 µg isolated ECM in 150 µL total volume was kept under constant agitation to avoid precipitation for 1 h. The collagen IV crosslinking was induced with 100 µM of H_2_O_2_ and 100 µM of NaBr in PBS buffer (pH 7.4).Then, reactions were stopped with 1% sodium azide. The non-collagenous domain (NC1) was obtained after digestion by collagenase 1 (Gibco, Life Technologies, Waltham, MA, USA) (2 mg collagenase/5 mg ECM) overnight. The expression of collagen IV NC1 dimers and monomers was verified by Western blot. Briefly, 100 µg of the experimental samples were separated on a 10% reducing SDS-polyacrylamide gel and transferred to a polyvinylidene difluoride (PVDF) membrane. Membranes were incubated with primary antibody to Anti-ColIVA2 (1:5000) and subsequently incubated with HRP-conjugated secondary antibody (anti-rabbit—1:10,000) to detect protein expression by chemiluminescence (Chemidoc, Bio-Rad Laboratories, Hercules, CA, USA). Band densitometry was calculated using ImageJ software.

### 2.9. Adhesion Assay

To evaluate the capability of cells to attach, confluent HUVECs were harvested with EDTA (0.5 mM in PBS) and distributed in microtubes (2.5 × 10^4^ cells/tube) containing modified RPMI medium with 0.2% BSA and the indicated treatments: PHG (10, 50, and 100 μM) or urate (25–400 μM). After gentle mixing, cells were seeded in non-coated or 1% gelatin coated 96-well plates and left to adhere for 24 h at 37 °C and 5% CO_2_. After incubation, the supernatant was removed and adherent cells were washed 3 times with ice-cold PBS, followed by a fixation step with methanol (10 min, −20 °C). After washing 3 times with distilled water, cells were stained with crystal violet (0.1% in 25% methanol), for 30 min at room temperature and washed again 3 times with distilled water. Crystal violet was solubilized with 1% SDS and the absorbance was read at 600 nm in a microplate reader (Biotek Synergy, Biotek, Winooski, VT, USA). Absorbance was normalized to the control group (100%). 

### 2.10. Wound Healing Assay

HUVECs were harvested and seeded in 6-well plates at an initial confluence of 5 × 10^5^ cells/well. After 2 days, the culture medium was replaced by RPMI containing 0.2% BSA for non-transfected cells. For transfected cells, a transfection medium was added after washing the cells with PBS [42]. After 24 h, a scratch was made in the cell monolayer. Cells were washed 3 times with PBS to remove debris and RPMI 0.2% BSA was added with the indicated treatments: PHG (10, 50 and 100 μM) or urate (25–400 μM). Reference marks were made in the plates, and the images were captured in an inverted light microscope (Nikon TE300, Minato, Tokyo, Japan). Cells were incubated for 24 h at 37 °C and 5% CO_2_ for 24 h and the images at the final incubation period were acquired. ImageJ software was used to measure the distance of each scratch using: 

%migration = (initial scratch width − final scratch width)/initial scratch width × 100%

### 2.11. SDS-PAGE Analysis

Fifty µg protein from HUVEC secretome was mixed with Laemmli buffer, heated at 95 °C for 10 min, and separated in 10% SDS-PAGE. The gels were stained with Coomassie blue G for 2 h and washed with water overnight.

### 2.12. Protein Identification by Proteomics

#### 2.12.1. In-Gel Digestion

For in-gel digestion, the bands corresponding to 100–250 kDa were excised from the gel, cut into 1 mm^3^ pieces, and destained with 50% acetonitrile in water. Samples were reduced with DTT (10 mM) and alkylated with iodoacetamide (50 mM) for 30 min. The reduced and alkylated protein samples were digested with Sequencing Grade Modified Trypsin (Promega - Madison, WI, USA) (protein/trypsin ratio of 50:1) in ammonium bicarbonate (50 mM) for 4 h. Then, a second aliquot of trypsin (protein/trypsin ratio 50:1 *w*/*w*) was added, and the protein samples were further incubated at 37 °C overnight. Digestion was stopped with 5% formic acid in 50% acetonitrile. Peptides were eluted from the gel fragments once with 1% formic acid in 60% methanol and twice with 60% acetonitrile containing 1% formic acid and dried under vacuum. Before analysis, the peptides were suspended in 0.1% formic acid in water and desalted using a Stage Tip protocol [43]. Samples were dried and stored at −80 °C until MS analyses. 

#### 2.12.2. In-Solution Digestion

First, 10 μg protein was solubilized in 100 mM ammonium bicarbonate with 0.4% sodium deoxycholate, reduced with DTT (5 mM), alkylated with iodoacetamide (15 mM), digested with PNGase F Glycan Cleavage Kit (Gibco-Life Technologies, Waltham, MA, USA) for 1 h at 37 °C and digested with Sequencing Grade Modified Trypsin (Promega-Madison, WI, USA), protein/trypsin ratio 40:1 *w*/*w* for 4 h at 37 °C. A second aliquot of trypsin (50:1 *w*/*w*) was added and samples were incubated overnight at 37 °C. After acidic hydrolysis with 2% trifluoroacetic acid, samples were desalted using the StageTip protocol [43]. 

#### 2.12.3. Extracellular Matrix (ECM) Digestion

ECM digestion from HUVEC, HEK293, and HEK293 overexpressing PXDN cells were prepared following the in-solution digestion described above but with the replacement of sodium deoxycholate by 8 M urea and 6 M guanidine hydrochloride in the solubilizing step [44].

#### 2.12.4. Data Dependent Acquisition (DDA) Proteomics

Digested and desalted samples were suspended in 0.1% formic acid (final protein concentration of 25 ng/μL) and submitted to MS analyses. Angiotensin (0.2 pmol/μL) was used as a global internal standard to monitor mass variability, and iRT peptides (Pierce Biotechnology, Rockford, IL, USA, 0.1 pmol/μL) were used to normalize the retention time of all peptides. An Easy-nLC 1200 UHPLC (Thermo Scientific, Bremen, Germany) was used for peptide separation with a linear gradient of solvent A (0.1% formic acid) and solvent B (0.1% formic acid in 80% acetonitrile). Each sample was loaded onto a trap column (nanoViper C18, 3 μm, 75 μm × 2 cm, Thermo Scientific, Bremen, Germany) with 12 μL of solvent A at 980 bar. After this period, the trapped peptides were eluted onto a C18 column (nanoViper C18, 2 μm, 75 μm × 15 cm, Thermo Scientific, Bremen, Germany) at a flow rate of 300 nL/min. Peptides were eluted from the column using a linear gradient of 5−28% B for 25 min followed by 28−40% B for 5 min. Finally, the percentage of solvent B was increased to 95% in 2 min, and the column was washed for 10 min with this solvent proportion. Re-equilibration of the system with 100% A was performed before each injection. Acquisition of the data was performed using an Orbitrap Fusion Lumos mass spectrometer (Thermo Scientific, Bremen, Germany) with a nanospray Flex NG ion source (Thermo Scientific, Bremen, Germany). A full MS scan was followed by data-dependent MS2 scans in a 3 s cycle time. Precursor ions selected for MS2 were excluded for subsequent MS2 scans for 20 s. The resolution for the full scan mode was set as 120,000 (at *m/z* 200) and the automatic gain control (AGC) target at 4 × 10^5^. The *m/z* range 400−1600 was monitored. Each full scan was followed by a data-dependent MS2 acquisition with a resolution of 30,000 (at *m/z* 200), maximum fill time of 54 ms, isolation window of 1.2 m/z, and normalized collision energy of 30.

#### 2.12.5. Protein Identification

Tandem mass (MS/MS) spectra were searched against the reviewed UniProt human database, using the MaxQuant search engine (Munich, Bavaria, Germany, www.maxquant.org accessed on 30 May 2022) with fixed Cys carbamidomethylation, variable Met oxidation, Asn-Asp exchange, and N-terminal acetylation. MaxQuant default mass tolerance was used for precursor and product ions. Trypsin/P was selected as the enzyme, and two missed cleavages were allowed. The results were processed by label-free quantification.

### 2.13. Statistical Analysis 

All data are expressed as mean ± standard error of the mean (SEM) of at least 3 independent experiments. The difference among groups was calculated by one-way ANOVA with Bonferroni’s post hoc test and was considered significant when *p* < 0.05. All analyses were performed using GraphPad Prism v.5 (GraphPad Software, San Diego, CA, USA, www.graphpad.com accessed on 30 May 2022).

## 3. Results

### 3.1. Urate Inhibits PXDN Released from HUVECs

Initially, peroxidase activities from HUVEC cellular fractions, secretome, lysate, and extracellular matrix (ECM) were compared. Secretome had the highest activity (Figure 1A), indicating that a peroxidase is being secreted by HUVECs. Proteins from secretome were separated by SDS-PAGE gel, and a band around 165 kDa was observed (Figure 1B). Proteomics analysis revealed the presence of PXDN unique peptides in this ~165 kDa gel band in the whole secretome and ECM from HUVEC (Appendix A). In presence of urate, there was a significant decrease in the oxidation of Amplex Red in HUVEC secretome. This decrease was comparable to other known heme-peroxidase inhibitors such as ABAH (4-aminobenzoic acid hydrazide) and PGH (phloroglucinol) (Figure 1C). Urate seems to preferentially inhibit PXDN since its incubation with PXDN knocking-down cells produced no further inhibition (Figure 1D). These results indicate that urate could be either a PXDN substrate or inhibitor. 

Peroxidase activity was also measured in the secretome and ECM from HEK293 overexpressing PXDN cells. HEK293/PXDN lysates, as well as in HUVEC, did not have considerable peroxidase activity. ECM fraction in these cells presented higher peroxidase activity than secretome (Figure 2A). In addition, in the ECM fraction from PXDN overexpressing cells the fluorescence was twice higher than in the secretome and lysate at zero time. This higher fluorescence could be due to the higher peroxidase activity and to the presence of trace hydrogen peroxide in this sample. Therefore, during the lag between placing the plate in the reader and the detection, a considerable amount of Amplex Red had already been oxidized. Oxidation of Amplex Red was potently prevented by urate (Figure 2B,C), and the inhibition was much more pronounced in the ECM fraction (Figure 2B) than in the secretome (Figure 2C). Since the concentration of urate was much lower than Amplex Red, the decrease in Amplex Red oxidation suggested that urate is a PXDN inhibitor rather than an alternative substrate.

In accordance, recent data using a truncated PXDN showed that urate reacts with compound I PXDN at 1.9 × 10^4^ M^−1^s^−1^ and with compound II at 5.8 × 10^2^ M^−1^s^−1^ (100 mM phosphate buffer, pH 7.4 at 25 °C). This near two orders of magnitude difference among rate constants holds the enzyme as the intermediate compound II, preventing the complete cycle and the return to its native form [34]. The rate constants were calculated from a truncated PXDN because of the limitations in the full-length recombinant protein purification. Thus, our study extends these previous findings because it reports the effect of urate on a full-length protein. 

In an attempt to understand why HUVEC has higher peroxidase activity in secretome, whereas HEK293 cells in the ECM fraction, we identified protein content from ECM in both HUVEC and HEK293 cells by proteomics. Laminin and type IV collagen are the largest class of proteins present in the basement membrane interacting with each other and with other ECM proteins. This laminin/type IV collagen colocalization consequently places PXDN. A recent study showed that PXDN binds to laminin in the basement membrane [45]. Thus, we wondered if laminin content from a specific type of cell could influence PXDN presence since we observed differences in PXDN activity using ECM, secretome, and lysate fractions from distinct cell lines. The intensities of laminin subunits unique peptides from HUVEC, HEK293, and HEK293 cells overexpressing PXDN ECM are highlighted in the Appendix A. Laminins are heterotrimers (αβγ) assembled by chains of genetic variants in different combinations, generating a variety of isoforms. Laminin 111 (α1β1γ1) isoform was the first one to be discovered, and it has been studied due to its role in basement membrane assemblage, contributing to cell adhesion and cell signaling [46]. Laminin γ1 subunit was identified in ECM from HUVEC presenting a lower LFQ intensity than in HEK293 and HEK293/PXDN overexpressing cells. HEK293 cells presented β1γ1 subunits, but the laminin α1β1γ1 heterotrimers were identified in HEK293/PXDN overexpressing cells only. These observations corroborate recent findings showing a PXDN dependence on the expression of collagen IV, fibronectin, and laminin in ECM [47]. Thus, PXDN presence might contribute to laminin production. However, further studies are needed for a certain conclusion.

### 3.2. Urate Decreases PXDN-Bromination Activity

The effect of urate on Amplex Red oxidation could be a non-specific effect upon any peroxidase present in ECM or secretome. However, bromination is a specific result of heme-peroxidases, such as PXDN, catalysis [28,39]. Hypobromous acid (HOBr) was indirectly measured through the oxidation of tyrosine to 3-bromotyrosine and 3,5-dibromotyrosine [40]. As expected, bromination activity was higher in ECM of HEK293/PXDN overexpressing cells than in HUVEC secretome (Figure 3). Urate (200 µM) and phloroglucinol (50 µM) inhibited bromination in both ECM from HEK293/PXDN overexpressing and secretome from HUVEC. These results demonstrated that urate can efficiently decrease HOBr and, therefore, could affect collagen-IV crosslinking. Since urate is also a HOBr scavenger, we cannot discard a direct reaction. However, since tyrosine is in a five-fold excess over uric acid, this decrease can be attributed, at least in part, to a diminution in HOBr production by the enzyme. These data are in agreement with those that demonstrated a decrease in bromination activity of the truncated form of PXDN by urate [34]. 

### 3.3. Identification of Urate Oxidation Products in HUVEC Secretome

Since urate can be a substrate for PXDN [34], we identified and quantified the products of this reaction. As previously reported by our group, the one-electron oxidation of urate by the heme-peroxidase, myeloperoxidase, generates the intermediate hydroxyisourate [19]. Hydroxyisourate was also the product of urate oxidation by PXDN (Figure 4). Oxidation of urate occurred even when exogenous H_2_O_2_ was omitted, demonstrating that endogenous H_2_O_2_ is enough for PXDN catalysis in HUVEC secretome. Of relevance, products of urate oxidation by heme-peroxidases can form adducts with lysine residues in proteins [36,48], which may alter protein function.

### 3.4. Urate Reduces Collagen IV Crosslinking

To evaluate the effect of urate on collagen IV crosslinking in isolated ECM, we initially treated PFHR9 cells with the heme-peroxidase inhibitor, PHG, for 6 days to prevent the formation of collagen IV crosslinks. As illustrated in Figure 5, dimer bands with one and two crosslinks were intensely marked in the control (37 kDa), but not in absence of bromide or PHG. Treatment with 20 to 200 µM urate decreased dimer bands. These results corroborate a previous study showing a 45% decrease in the formation of collagen IV dimers in presence of 20 µM urate [39]. 

### 3.5. PXDN Inhibition Decreases HUVEC Migration and Adhesion

To access whether PXDN inhibition would affect HUVEC migration, we performed an in vitro scratch assay [42]. The treatment with 200 and 400 μM urate and 50 and 100 μM PHG significantly decreased cell migration (Figure 6A,B). To evaluate a detachment effect, previously seeded and confluent HUVECs were treated with urate (10–400 μM) and PHG (10–100 μM). After 24 h of treatment, cells were gently washed with PBS. No difference was observed in this experiment (data not shown). Then, we evaluated cell adhesion from a different standpoint, in which HUVEC cells were harvested and treated with urate (10–400 μM) and PHG (10–100 μM) while seeding for 24 h. Both urate (400 μM) and PHG (50–100 μM) decreased cellular adhesion by approximately 30% when cultivated in plates without gelatin coating, but there was no difference in cell adhesion with gelatin coating (Figure 6C–D). Lastly, we evaluated migration and adhesion in siRNA PXDN HUVEC transfected cells. When compared to siRNA scramble transfected cells, PXDN silencing decreased HUVEC migration and adhesion in the non-coated surface by approximately 10% and 40%, respectively (Figure 7A–C). 

## 4. Discussion

The discovery of PXDN catalysis in sulfilimine bond formation was a breakthrough event to unveil the mechanism of collagen IV crosslink. It was also the first evidence of bromine’s role in mammals [33]. However, it remains unclear whether the physiological function of PXDN is restricted to sulfilimine-bond catalysis or not. PXDN has been correlated with angiotensin II-mediated cell proliferation [49] and angiogenesis promotion [50]. It is essential for endothelial cell survival and extracellular matrix structuring [47]. The enzyme was also associated with the progression of prostate cancer [51], ovarian cancer [52], melanoma cellular invasion [53], and kidney fibrosis [54]. Of relevance, aberrant PXDN expression has been associated with cardiovascular disorders, including endothelial dysfunction in hypertension [55,56], type 2 diabetes [57], and atherosclerosis [58]. 

By using a DDA proteomics approach that included an extra digestion step with PNGase, we identified 22 unique peptides for PXDN in HUVEC ECM and secretome. One of these peptides, the AGEIFER, was also identified in a previous study that used MS to quantify PXDN in cellular extracts [59]. Additionally, our results corroborate previous Western blot data that demonstrated PXDN secretion by HUVEC [60]. By confirming the presence of extracellular PXDN in HUVEC, we decided to further investigate the role of PXDN in endothelial cells using this cell line. 

The heme-peroxidases from the PXDN family, myeloperoxidase and lactoperoxidase, are known to oxidize the endogenous substrate uric acid, yielding urate free radical and urate hydroperoxide [18,19,20]. Urate free radical can initiate a free radical chain reaction, whereas urate hydroperoxide oxidizes thiols from peptides and proteins [21,22,61]. Recently, we published the first clinical study correlating urate oxidation with atherosclerosis. As a part of a Brazilian cohort study, we found a significant association between the levels of uric acid and the end product of its oxidation, allantoin, with carotid intima-media thickness (c-IMT). Patients with c-IMT higher than 1 mm were excluded from the investigation to assure that an initial process of atherosclerosis was taking place. Classical atherosclerosis risk factors such as LDL/HDL ratio, neck circumference, and pulse pressure did not correlate with c-IMT in these patients. Additionally, the correlation of uric acid and allantoin with c-IMT was independent of the above-mentioned risk factors, indicating that uric acid oxidation is occurring at the beginning of the atherosclerotic process and may have a causal role in disease progression. Of note, no positive correlation between allantoin and myeloperoxidase was found, indicating that, in this initial condition, the other two electron oxidants and other vascular peroxidases, including PXDN, might be leading to uric acid oxidation [11]. 

In agreement with the oxidation of urate by PXDN, Sevcnikar and collaborators demonstrated that urate is a good substrate for the truncated PXDN intermediate compound I (1.9 × 10^4^ M^−1^s^−1^), but it is a poor substrate to the compound II (5.8 × 10^2^ M^−1^s^−1^) [34]. Therefore, at physiological concentrations, urate can compete with bromide by compound I, and in the absence of other compound II substrates, it slows down the enzyme turnover. In addition, when using isolated ECM from PFHR9 cells, urate decreased collagen IV crosslink [39]. Here, we also observed a decrease in collagen IV crosslink by urate in isolated ECM. However, no inhibition was found when urate was added to the culture medium for 6 days on seeded PFHR9 cells (data not shown). In this case, it is possible that urate is not accessing the enzyme, and therefore, a different location of the enzyme in the different tissues will determine whether urate will be or not oxidized by the enzyme. 

Whereas HEK293/PXDN overexpressing cells had the highest peroxidase activity in ECM enriched fraction, HUVEC had it in secretome. PXDN interacts with laminin in the ECM [45], and we wondered if laminin content would be different in these two types of cells and whether it could influence enzyme retention in the basement membrane. Proteomic analyses revealed that the laminin gamma 1 subunit is much lower in HUVEC ECM than in HEK293 cells. In addition, alpha 1 and beta 1 subunits were not found in HUVEC (Appendix A). These findings suggest that the presence of these laminin subunits is necessary for PXDN attachment to ECM. However, more studies are needed to confirm this correlation.

We also showed that urate decreases the brominating activity of PXDN. As a result, there was a decrease in the formation of sulfilimine crosslinks. The decrease in tyrosine bromination could be by the competition between urate and bromide by PXDN compound I, as well as by a direct reaction of urate with hypobromous acid. In the former, the one-electron oxidation forms urate radical. Two urate radicals can dismutate yielding dehydrourate and urate. In the latter, the two-electron oxidation of urate also forms dehydrourate [19]. This common product is hydrated to hydroxyisourate, the oxidized product that we identified in HUVEC secretome. Since our bromo-tyrosine assay contains an excess of tyrosine over urate, it suggests that the decrease in tyrosine bromination is due, at least in part, to the competition of urate by PXDN compound I. Urate oxidation by PXDN compound I is relevant because urate free radical can trigger a free radical chain reaction and oxidative stress.

Incubation with urate decreased HUVEC migration and adhesion. Both events are essential for tissue repair, angiogenesis, and vascular integrity maintenance [62,63] and might be a consequence regarding the decrease in PXDN bromination activity. In accordance, the heme peroxidase inhibitor, PHG, and PXDN silencing also inhibited HUVEC migration and adhesion. In agreement with our findings, previous studies demonstrated that PXDN activity is important for endothelial cell migration, tubulogenesis, and proliferation [47,50]. 

Hyperuricemia has been extensively studied for triggering inflammation and endothelial dysfunction [14,64,65,66,67,68]. However, studies that evaluate the effects of physiological concentrations of urate upon endothelial cell function are scarce. A physiological concentration of uric acid (300 µM) did not affect the proliferation of endothelial cells induced by EGF. However, a hyperuricemic concentration, 600 µM, leads cells to senescence [69] and inhibition of migration [70]. Higher concentrations (720 µM) induced ER stress [71,72]. 

Oxidation of urate is also relevant in endothelial cell adhesion impairment. Our group recently showed that the product of urate oxidation, urate hydroperoxide, disrupted HUVEC adhesion by oxidizing thiol proteins that are key to cell adhesion [21]. However, urate hydroperoxide production is mostly dependent on superoxide. Therefore, although the oxidation of urate by PXDN and production of urate free radical may occur in the endothelium, the production of urate hydroperoxide and selective oxidation of thiols are limited to superoxide presence. Thus, the effects of uric acid on cell adhesion and migration may be mostly due to the decrease in hyprobromous formation and to an unspecific reaction of urate free radical with proteins. 

## 5. Conclusions

In summary, this study provides additional information on the role of PXDN in ECM formation. It shows, for the first time, that inhibition of PXDN is enough to impair adhesion and migration of vascular endothelial cells in absence of another injury. The purine metabolite, urate, can be oxidized by PXDN, producing the unstable urate free radical, decreasing hypobromous acid formation and disrupting ECM formation and HUVEC adhesion and migration. Therefore, this study supports a harmful effect of physiological concentrations of uric acid upon vascular endothelial cells, reinforcing the relevance of this metabolite in cardiovascular disease.

## Figures and Tables

**Figure 1 antioxidants-11-01117-f001:**
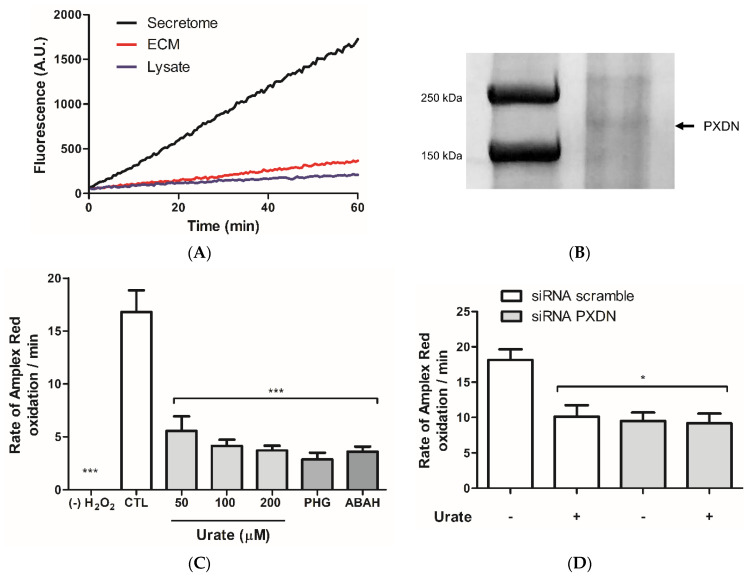
Urate decreases Amplex Red oxidation in HUVEC secretome. (**A**) Peroxidase activity in HUVEC secretome, extracellular matrix (ECM), and lysate. Representative data from two independent experiments. HUVECs were cultured in RPMI medium (10% FBS). After total confluence, cells were incubated with a modified RPMI medium without phenol red and FBS for 90 min, and the secretome was concentrated (see Methods). Peroxidase activity was measured by reacting 50 µM Amplex red plus 50 µM H_2_O_2_ and 10 µg protein in 50 mM phosphate buffer, pH 7.4. (**B**) Coomassie-brilliant-blue-stained SDS-PAGE gel from HUVEC secretome. A band between 250 and 150 kDa was selected and PXDN was identified by proteomics (Appendix A). (**C**) Urate or known heme-peroxidase inhibitors prevented Amplex Red oxidation in HUVEC secretome. (**D**) Urate (200 µM) prevented Amplex Red oxidation similarly to siRNA PXDN. HUVECs were treated with 50 nM PXDN siRNA or scramble (control). After 24 h medium was changed to a RPMI without phenol red and FBS for 90 min, and the secretome was concentrated. Bar graphs represent the mean ± S.E.M. (n = 3). Statistical analyses were performed by one-way ANOVA followed by Bonferroni’s post-test, *** *p* < 0.001 and * *p* < 0.05 when compared to control or siRNA scramble. (-): absence; PHG (50 µM): phloroglucinol; ABAH (50 μM): aminobenzoic acid hydrazide.

**Figure 2 antioxidants-11-01117-f002:**
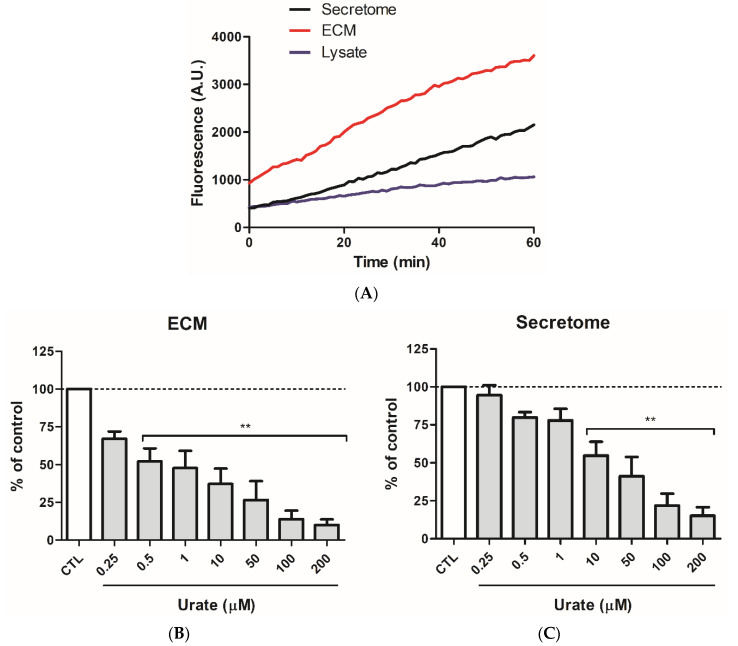
Urate potently decreases Amplex Red oxidation in HEK293/PXDN overexpressing cells. (**A**) Peroxidase activity in HEK293/PXDN secretome, extracellular matrix (ECM), and lysate. Urate prevented Amplex Red oxidation in (**B**) ECM and (**C**) secretome from HEK293/PXDN overexpressing cells. HEK293/PXDN cells were cultured in DMEM High Glucose medium (10% FBS). After total confluence, cells were incubated with DMEM High Glucose F12 medium without phenol red and FBS for 36 h. ECM was isolated and secretome was concentrated. Peroxidase activity in 10 µg total protein was measured by incubating it with 50 µM amplex red plus 50 µM H_2_O_2_ in 50 mM phosphate buffer, pH 7.4. Each bar represents the mean ± S.E.M of three independent experiments. Statistical analyses were performed by one-way ANOVA followed by Bonferroni’s post-test. ** *p* < 0.01 compared to control (CTL, absence of urate).

**Figure 3 antioxidants-11-01117-f003:**
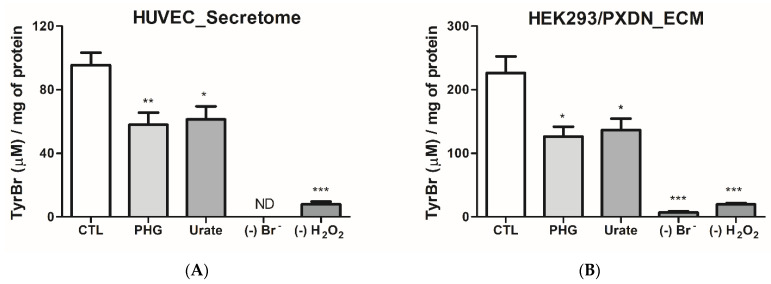
Urate decreases tyrosine bromination by PXDN. (**A**) Secretome from HUVEC and (**B**) extracellular matrix (ECM) from HEK293/PXDN overexpressing cells, which were incubated with tyrosine (1 mM), H_2_O_2_ (100 µM) and NaBr (100 mM) (CTL group), urate (200 µM) or PHG (50 µM) in 50 mM phosphate buffer (pH 7.4) for 4 h at 37 °C before measurement of 3-bromotyrosine or 3,5-dibromotyrosine by HPLC. Each datum represents the mean ± S.E.M. (n = 3). Statistical analyses were performed by one-way ANOVA followed by Bonferroni’s post-test, * *p* < 0.05 ** *p* < 0.01 *** *p* < 0.001 compared to the control group. (-) Br^−^: no Br^−^ addition; (-) H_2_O_2_: no H_2_O_2_ addition; PHG: phloroglucinol.

**Figure 4 antioxidants-11-01117-f004:**
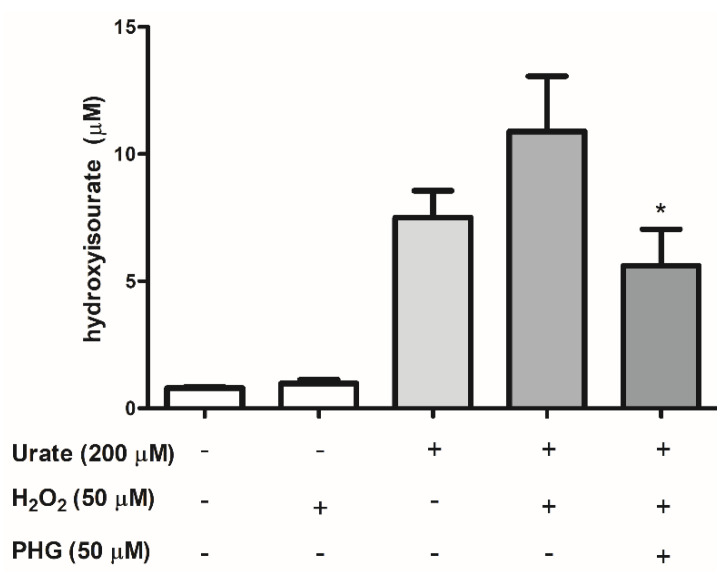
Oxidation of urate by PXDN produces 5-hydroxyisourate. HUVEC secretome was concentrated and incubated with 200 µM urate in the absence or presence of 50 µM H_2_O_2_ and 50 µM PHG in 50 mM phosphate buffer, pH 7.4 for 30 min at 37 °C. After incubation, samples were suspended in 2% trichloroacetic acid (TCA) and 60% acetonitrile (ACN), centrifuged and the supernatant was injected onto LC/MS/MS. Each datum represents the mean ± S.E.M. (n = 3). Statistical analyses were performed by one-way ANOVA followed by Bonferroni’s post-test, * *p* < 0.05 compared to the control group. PHG: phloroglucinol.

**Figure 5 antioxidants-11-01117-f005:**
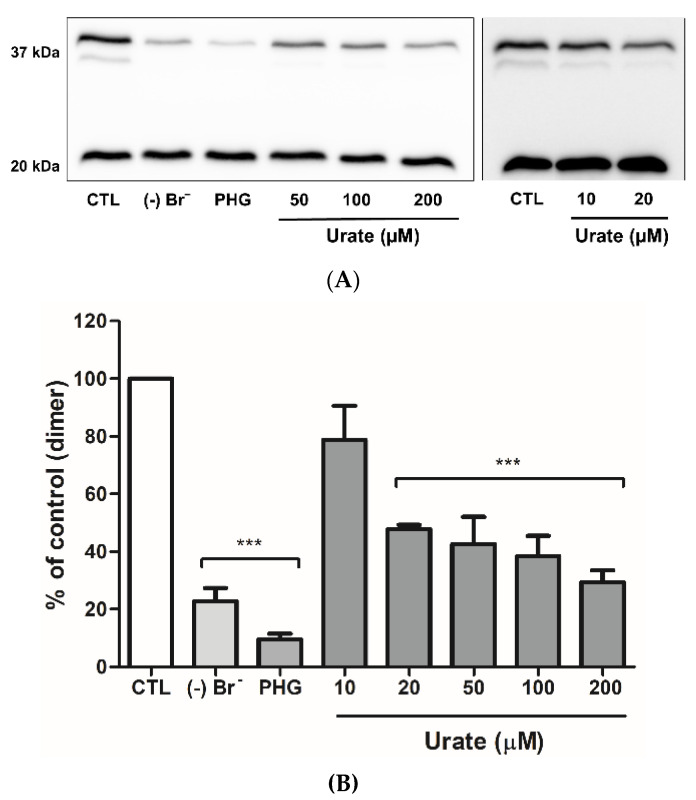
Urate inhibits collagen IV crosslinking in isolated ECM from PFHR9 cells. After achieving confluence, PFHR9 cells were cultured in a DMEM high glucose medium containing 50 µM PHG for 6 days. Then, ECM was isolated, and 750 µg was used in each reaction. The collagen IV crosslinking was induced with 100 µM of H_2_O_2_ and 100 µM of NaBr (CTL group) for 1 h. (**A**) Collagen IV crosslink dimers, around 37 kDa; upper band with one sulfilimine bond and lower band with two sulfimine bonds and non-crosslinked collagen monomers, around 20 kDa, were visualized by ColIV4α2 immunoblot. The evaluation of 10 and 20 µM of urate (right panel) was performed in different days. (**B**) Dimer densitometry related to the respective control of each experiment. Each datum represents the mean ± S.E.M. (n = 3). Statistical analyses were performed by one-way ANOVA followed by Bonferroni’s post-test, *** *p* < 0.001 compared to the control group. (-) Br^−^: no Br^−^ addition; PHG (50 µM): phloroglucinol.

**Figure 6 antioxidants-11-01117-f006:**
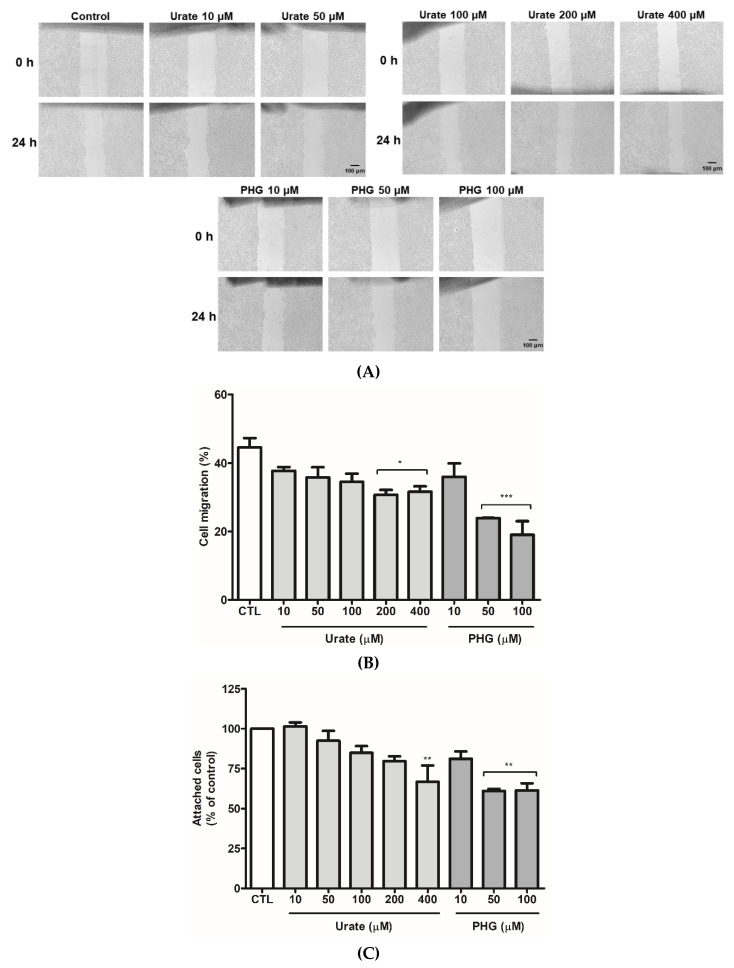
Urate and PHG reduce HUVEC migration (**A**,**B**) and adhesion (**C**,**D**). (**A**,**B**) HUVECs (5 × 10^5^ cells/well) were seeded in 6-well plates. After 2 days, the culture medium was replaced by RPMI medium 0.2% BSA. After 24 h, a scratch was made in the cell monolayer. Cells were washed with PBS and incubated with RPMI medium 0.2% BSA (control), urate (10–400 μM), and PHG (10–100 μM) for 24 h. (**A**) Images of scratch were taken in an inverted light microscope (Nikon) using 4× magnification before and after incubation. Scale bar 100 μm. Representative images of 3 independent experiments. (**B**) The distance of the scratch was calculated using ImageJ software. The results are expressed as %migration = (initial scratch width − final scratch width)/initial scratch width *100%**.** (**C**,**D**) HUVECs (2.5 × 10^5^ cells/well) were harvested and incubated with RPMI medium 0.2% BSA (control), urate (10–400 μM), and PHG (10–100 μM) while seeding for 24 h in 96-well plates without coating (**C**) or with gelatin coating (**D**). After incubation, cells were washed with PBS and stained with crystal violet. Each datum represents the mean ± S.E.M. (n = 3). Statistical analyses were performed by one-way ANOVA followed by Bonferroni’s post-test, * *p* < 0.05 ** *p* < 0.01 *** *p* < 0.001 compared to the control group. PHG: phloroglucinol.

**Figure 7 antioxidants-11-01117-f007:**
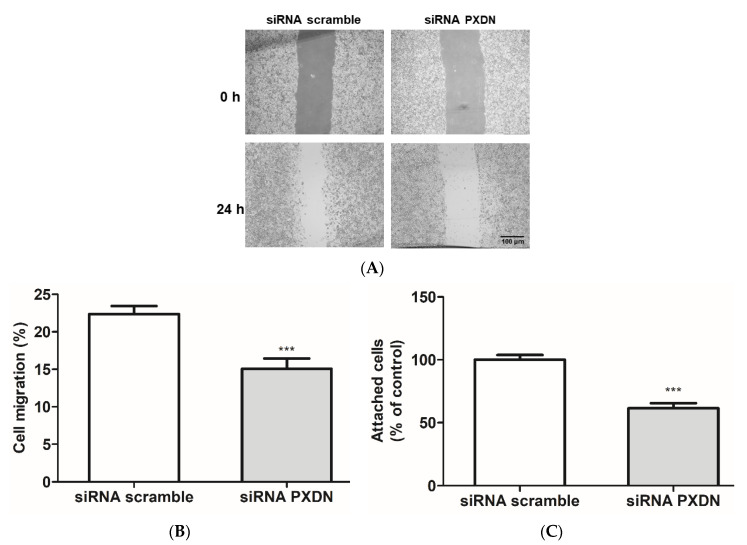
PXDN silencing reduces migration and adhesion of HUVECs. (**A**,**B**) HUVECs (5 × 10^5^ cells/well) were seeded in 6-well plates. After 2 days, the culture medium was replaced by a transfection medium containing 50 nM of siRNA. Twenty-four hours after transfection, a scratch was made in the cell monolayer, and cells were washed with PBS and incubated with RPMI medium 0.2% BSA for 24 h. (**A**) Images of scratch were taken in an inverted light microscope (Nikon) using 5× magnification before and after the incubation period. Scale bar 100 μm. Representative images of 3 independent experiments. (**B**) The distance of the scratch was calculated using ImageJ software. (**C**) HUVECs were transfected with siRNA for 24 h. After the incubation period, cells were harvested and seeded (2.5 × 10^5^ cells/well) for 24 h in 96-well plates without coating in RPMI medium 0.2% BSA. Cells were then washed with PBS and stained with crystal violet. Each datum represents the mean ± S.E.M. (n = 3). Statistical analyses were performed by one-way ANOVA followed by Bonferroni’s post-test, *** *p* < 0.001 compared to the control group.

## Data Availability

The data presented in this study are available on request from the corresponding author. The data are not publicly available due to rights on University of Sao Paulo and grants agency FAPESP.

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
