# Peer review of "Uric Acid Reacts with Peroxidasin, Decreases Collagen IV Crosslink, Impairs Human Endothelial Cell Migration and Adhesion"

_antioxidants, 2022, doi:10.3390/antiox11061117_

Round 1

Reviewer 1 Report

Peroxidasin (PXDN) is an animal heme peroxidase family with a crucial biological function in basal membrane synthesis. The enzyme oxidizes bromide into hypobromous acid (HOBr) to crosslink collagen IV,  through the formation of sulfinimine links. This activity of the enzyme has been published in high-profile papers (including Nature Chemical Biology, Cell) and these findings represent the most fascinating developments in the field. Although PXDN has a crucial role in ECM formation it is possible that PXDN’s activity is not restricted to collagen IV crosslinking.

The authors of this manuscript studied the effect of uric acid on the collagen IV crosslinking activity of PXDN. Although uric acid was previously described as a substrate of PXDN, the authors put this relationship into physiological context, thus their findings are novel and important. They also demonstrate that uric acid (at physiological concentrations) inhibits the collagen IV crosslinking activity of PXDN, therefore it is possible that this metabolite affects extracellular matrix formation by endothelial cells.

I have the following comments:

1. Uric acid seems to alter collagen IV crosslinking at physiological concentrations. Is it possible that urate oxidation by PXDN represents a physiological function of the enzyme? This possibility becomes especially interesting if PXDN is secreted into a compartment/location, where collagen IV is not present.

2. In Figure S1 a loading control should be included.

3. Figure 2: The authors should comment on the increased fluorescence value observed at 0 min when the peroxidase activity of HEK293-derived ECM was assessed.

4. In Figure 5 the authors show the effect of increasing urate concentrations on collagen IV crosslinking. What is the difference between the two panels where the effect of increasing urate concentrations is demonstrated? In this figure label, “B” should be moved closer to the densitometry data.

5. In Figure 5 the inhibitory effect of bromide depletion on collagen IV crosslinking is shown. How did the authors remove bromide from the cell culture medium?

6. What do the numbers (%) represent in Figure 6B?

Author Response

We thank the reviewers for taking time to review this manuscript, please observe that d we have considered all the comments that they made. We are sure that these changes have improved the quality of the manuscript.

Response to Reviewer #1:

Minor concerns:

1) Uric acid seems to alter collagen IV crosslinking at physiological concentrations. Is it possible that urate oxidation by PXDN represents a physiological function of the enzyme? This possibility becomes especially interesting if PXDN is secreted into a compartment/location, where collagen IV is not present.

Response: Yes, this was our first hypothesis. Considering plasma concentration of urate versus bromide and the rate constants of the reaction with these two substrates, urate oxidation may occur physiologically mainly when the enzyme is secreted. We observed inhibition of Amplex Red oxidation by urate in secretome and ECM and inhibition of crosslink in isolated ECM. However, no crosslink inhibition was observed when urate was incubated with PFHR9 cells for 6 days. Therefore, urate is likely not accessing the enzyme in this PFHR9 cells culture. This information is now discussed in the manuscript (page 28).

2) In Figure S1 a loading control should be included.

Response: Please observe that we have added the loading control in this figure

3) Figure 2: The authors should comment on the increased fluorescence value observed at 0 min when the peroxidase activity of HEK293-derived ECM was assessed.

Response:  Please observe that a statement explaining it was added to the results section (page 15-16). “This higher fluorescence could be due to the higher peroxidase activity and the presence of trace hydrogen peroxide in this sample. Therefore, during the lag between placing the plate in the reader and the detection, a considerable amount of Amplex Red had already been oxidized.”

4) In Figure 5 the authors show the effect of increasing urate concentrations on collagen IV crosslinking. What is the difference between the two panels where the effect of increasing urate concentrations is demonstrated? In this figure label, “B” should be moved closer to the densitometry data.

Response: The panel containing 10 and 20 μM urate is separated from the other because it was performed in different days. It is now stated in the legend. We set the labels so all are lined with the y axis title. However, labels are not within the figure and they can be moved by the editorial office.

5) In Figure 5 the inhibitory effect of bromide depletion on collagen IV crosslinking is shown. How did the authors remove bromide from the cell culture medium?

Response: To assess the crosslinking, the isolation of ECM requires several washes. Therefore, bromide from culture media is removed and the reaction is performed in PBS. However, some trace bromide may be present in the buffer/washed ECM and, therefore, (-)Br- means no addition of bromide. This information was added in the legend.    

6) What do the numbers (%) represent in Figure 6B?

Response: the percentage in this figure is referent to cell migration and it was calculated as, %migration = (initial scratch witdh – final scratch witdh)/initial scratch witdh *100%. Please observe that we changed the statement in the methods section and added this information in the figure legend, as well.

Reviewer 2 Report

Effects of uric acid on peroxidasin (PXDN)-mediated reactions were studied on different cell lines. Uric acid is oxidized by peroxidasin to hydroxyisourate. This reaction inhibits the peroxidase activity of PXDN as measured by Amplex red, and the PXDN-mediated incorporation of bromine into tyrosine. Uric acid attenuates also collagen IV crosslinking and HUVEC migration and adhesion.

I have the following remarks.

Section 3.3 and Figure 5: At 37 kDa, two bands are visible. Are they the bands with one and two crosslinks? Please specify. Specify also the 20 kDa band. In the figure, the letter B should be moved to the right.

Line 340:  please indicate conditions (pH and temperature), under which the reaction rate constants were measured

Please include references for one-electron reduction potential of urate (line 32), and the extinction coefficient of HOBr (line 109).

Author Response

We thank the reviewers for taking time to review this manuscript, please observe that d we have considered all the comments that they made. We are sure that these changes have improved the quality of the manuscript.

Response to Reviewer #2:

1) Section 3.3 and Figure 5: At 37 kDa, two bands are visible. Are they the bands with one and two crosslinks? Please specify. Specify also the 20 kDa band. In the figure, the letter B should be moved to the right.

Response: Please observe that we have specified it in the legend figure “collagen IV crosslink dimers, around 37 kDa, upper band with one sulfilimine bond and lower band with two sulfimine bonds and non-crosslinked collagen monomers, around 20 kDa”. We set the labels so all are lined with the y axis title. However, labels are not within the figure and they can be moved by the editorial office

2) Line 340:  please indicate conditions (pH and temperature), under which the reaction rate constants were measured

Response: Please observe that we have added this information.

3) Please include references for one-electron reduction potential of urate (line 32), and the extinction coefficient of HOBr (line 109).

Response: Please observe that we have added this reference and information.